# Comparison of Sb_2_O_3_ and Sb_2_O_3_/SiO_2_ Double Stacked pH Sensing Membrane Applied in Electrolyte-Insulator-Semiconductor Structure

**DOI:** 10.3390/membranes12080734

**Published:** 2022-07-26

**Authors:** Chyuan-Haur Kao, Kuan-Lin Chen, Hui-Ru Wu, Yu-Chin Cheng, Cheng-Shan Chen, Shih-Ming Chen, Ming-Ling Lee, Hsiang Chen

**Affiliations:** 1Department of Electronic Engineering, Chang Gung University, 259 Wen-Hwa 1st Road, Kwei-Shan, Taoyuan 333, Taiwan; chkao@mail.cgu.edu.tw (C.-H.K.); chkao@mail.cgu.edu (K.-L.C.); 2Kidney Research Center, Department of Nephrology, Chang Gung Memorial Hospital, Chang Gung University, No.5, Fuxing St., Guishan, Taoyuan 333, Taiwan; 3Department of Electronic Engineering, Ming Chi University of Technology, 284 Gungjuan Rd., Taishan Dist., New Taipei City 24301, Taiwan; 4Center for Green Technology, Chang Gung University, 259 Wen-Hwa 1st Road, Kwei-Shan, Taoyuan 333, Taiwan; 5Department of Applied Materials and Optoelectronic Engineering, National Chi Nan University, Puli 545, Taiwan; s109328037@mail1.ncnu.edu.tw (H.-R.W.); s109328027@mail1.ncnu.edu.tw (Y.-C.C.); s107328012@mail1.ncnu.edu.tw (C.-S.C.); s107328009@mail1.ncnu.edu.tw (S.-M.C.); 6Department of Electro-Optical Engineering, Minghsin University of Science and Technology, No.1, Xinxing Rd., Xinfeng, Hsinchu 304, Taiwan

**Keywords:** Sb_2_O_3_/SiO_2_ double stack, pH sensing, silicate, crystallization, reiability

## Abstract

In this study, electrolyte-insulator-semiconductor (EIS) capacitors with Sb_2_O_3_/SiO_2_ double stacked sensing membranes were fabricated with pH sensing capability. The results indicate that Sb_2_O_3_/SiO_2_ double stacked membranes with appropriate annealing had better material quality and sensing performance than Sb_2_O_3_ membranes did. To investigate the influence of double stack and annealing, multiple material characterizations and sensing measurements on membranes including of X-ray diffraction (XRD), X-ray photoelectron spectroscopy (XPS), and scanning electron microscopy (SEM) were conducted. These analyses indicate that double stack could enhance crystallization and grainization, which reinforced the surface sites on the membrane. Therefore, the sensing capability could be enhanced, Sb_2_O_3_/SiO_2_-based with appropriate annealing show promises for future industrial ion sensing devices.

## 1. Introduction

In the 1980s, an enzyme sensor based on an electrolytic insulator semiconductor [1,2] (EIS) capacitive structure was demonstrated, which had a very simple structure. These EIS sensors consist of a dielectric layer [3] and are deposited on a silicon substrate [4]. The EIS sensor is immersed in a sample solution to form a capacitance structure and the capacitance of the structure changes [5] depending on the pH value of the contact liquid [6]. As the pH value changes, the flat band capacitance [7] of the EIS chip moves along the voltage axis [8]. Through the change of capacitance [9], the corresponding voltage change [10] can be measured according to the pH value (voltage-capacitance method). In recent years, several kinds of high-k dielectrics [11] have been proposed as sensing membranes such as WO_3_ [12], Y_2_O_3_ [13], Pr_2_O3 [14], and HfO_2_ [15] to substitute SiO_2_ [16] as a sensing membrane due to its low cost, compatibly with silicon, and compact size [17]. However, the dangling bonds and traps in high-k materials may cause great trouble in the future, and these peak applications in sensing devices [18]. SiO_2_ as the sensing membrane still has some advantages such as better crystallization and lower defect density [19]. Therefore, it is possible to have dual advantages by combing SiO_2_ and other dielectrics. In this work, double layer Sb_2_O_3/_SiO_2_ [20] sensing films were fabricated. Post-RTA in different temperatures (400 °C, 500 °C, 600 °C) was applied in EIS structure [21]. The sensor performance including sensitivity [22], hysteresis [23], and drift, rate [24], were measured to find the optimal annealing condition [25]. To examine the improvements of material properties, X-ray diffraction (XRD), X-ray photoelectron spectroscopy (XPS), and atomic force microscope (AFM) [26,27] were used to examine the crystalline structure, chemical bindings, and surface roughness. The results indicate that double stack and annealing could enhance crystallization and suppress silicate-related defects [28,29]. 

## 2. Experimental

The EIS structures were fabricated on 4-in n-type (100) Si wafers, which have a resistivity of 5–10 Ω-cm. The standard Radio Corporation of America (RCA) cleaning process was performed on the chips. The chip was first immersed in ethanol for 5 min and soaked in DI water for 5 min. Followed by immersion in acetone for 5 min, cleansed in DI water for 5 min, and then isopropyl alcohol was used to remove the organic contaminates. Then, the samples were dipped into 1% hydrofluoric acid to remove native oxide from the surface. A SiO_2_ film of 50 nm in thickness was grown on the wafer by dry thermal oxidation. Then, a 50-nm Sb_2_O_3_ was deposited by radio frequency (RF) reactive sputtering on an n-type silicon wafer and a SiO_2_ deposited wafer, respectively. During the RF sputtering, the Sb_2_O_3_ target was used in the ambient of Ar: O_2_ at 20:5. The RF power and pressure were 100 Watt and 20mTorr, respectively. Next, the samples were annealed at different temperatures (400 °C, 500 °C, and 600 °C) by rapid thermal annealing (RTA) in O_2_ ambient for 30 s. The back-side contact of the Si wafer was deposited by Al film with 300 nm-thick. Then, the sensing area was defined by a standard photolithography process using a photosensitive epoxy, SU8-2005 of Micro-Chem Inc. (Adel, GA, USA). Finally, the samples were fabricated on the copper lines of the printed circuit board (PCB) by silver gel. As for the sensing dielectric dimensions, the SiO_2_ film of 50 nm in thickness was grown on the wafer by thermal dry oxidation. Then, a 50-nm Sb_2_O_3_ was deposited by radio frequency (rf) reactive sputtering on the SiO_2_ film. The sensing area is 0.01 cm^2^. The oxide thicknesses were determined by high-frequency (100 kHz) capacitance-voltage (C-V) measurements. The expected error range of the oxide thicknesses for values is about 5–10%. An epoxy package was used to separate the EIS structure and the copper line. The detailed Sb_2_O_3_/SiO_2_ EIS structure is illustrated in Figure 1.

Figure 2a,b show the XRD patterns of the Sb_2_O_3_ layer and Sb_2_O_3_/SiO_2_ layer with various rapid thermal annealing conditions in O_2_ treatment. Obviously, in Figure 2a,b three diffraction peaks (222), (400) and (440) can be observed at 27.67°, 32.08°, and 45.96° can be observed for the two types of samples, respectively. When the temperature increased to 500 °C, the peak (222), (400) and (440) intensity increased. Due to enhancement of the crystalline structure during annealing, the peak intensity increased. The two types of samples annealed at a temperature of 500 °C exhibited the strongest peak of (222) and (440) among all of the samples. Moreover, compared with the single-layer (Sb_2_O_3_), the double-layer structure had a narrow full width at half maximum (FWHM), indicating the grain was larger. In addition, the peaks of the double stacked samples were stronger indicative of strong crystallization. The results reveal that the double-layer (Sb_2_O_3_/SiO_2_) had better crystallization and lower defect density.

To investigate the chemical-binding states in the Sb_2_O_3_/SiO_2_ and Sb_2_O_3_ sensing membranes, XPS was used. The O 1s spectra for the as-deposited Sb_2_O_3_ and Sb_2_O_3_ annealed films are shown in Figure 3a with their appropriate three-peak curve-fitting lines. In the three sets of spectra, the O 1s signal comprised three peaks at 532.9, 531.2 and 530.6 eV, which are SiO_2_, Sb-silicate, and Sb_2_O_3_ respectively. Compared to the single-layer (Sb_2_O_3_), the intensity of the double-layer (Sb_2_O_3_/SiO_2_) film had a stronger Sb_2_O_3_ peak and weaker silicate peak, especially for the sample annealed at 500 °C. This was due to the fact that the SiO_2_ layer can block Sb atoms from diffusion into the silicon to form silicate. After post-RTA annealing treatment at 500 °C in O_2_, the O 1s peak corresponding to Sb–silicate became weaker for the two types of samples indicative of improvement of crystallization and low defect density.

Furthermore, AFM was used to analyze the surface roughness. shown in Figure 4. The root means square (RMS) values of the Sb_2_O_3_ films and for the as-dep sample and the samples with RTA annealing from 400 °C to 600 °C were 0.250 nm, 0.267 nm, 0.292 nm, and 0.216 nm while the root means square (RMS) values of the Sb_2_O_3_/SiO_2_ for the as-dep sample and the samples with RTA annealing from 400 °C to 600 °C were 0.261 nm, 0.276 nm, 0.311 nm, and 0.242 nm. AFM was used to analyze the grainization effect of films. Consistent with XRD analysis, the double-layer structure has a narrower full width at half maximum, larger grains can make the surface roughness higher. In the case of 500 °C, there was the optimal lattice structure. As the annealing temperature approached the melting point, the crystal grains gradually disappear and the roughness began to decrease. When the temperature reached 600 °C, the roughness value decreased. This may be due to the fact there were many defects in the film and the lattice structure was destroyed.

To calculate the sensitivity and linearity of the Sb_2_O_3_ and Sb_2_O_3_/SiO_2_, C-V curves of the samples with various annealing conditions were graphed. The sensitivity values of the as-dep Sb_2_O_3_ film and the Sb_2_O_3_ films annealed at 400 °C, 500 °C and 600 °C were 42.25 mV/pH, 45.24 mV/pH, 51.96 mV/pH, and 34.44 mV/pH, respectively. The linearly values of the above four samples were 96.26%, 99.24%, 98.10%, and 82.69%, respectively. The sensitivity values of the as-dep Sb_2_O_3_/SiO_2_, film and the Sb_2_O_3_/SiO_2_, films annealed at 400 °C, 500 °C and 600 °C were 49.42 mV/pH, 49.77 mV/pH, 54.01 mV/pH, and 38.80 mV/pH, respectively. The linearly values of the above four samples were 99.41%, 98.89%, 99.03%, and 95.42%, respectively. 

Figure 5a–d show the C-V curves of the as-dep Sb_2_O_3_ film_,_ the Sb_2_O_3_ films annealed at 500 °C the as-dep Sb_2_O_3_/SiO_2_, film and the Sb_2_O_3_/SiO_2_, film annealed at 500 °C. Consistent with material characterizations, the Sb_2_O_3_/SiO_2_ sample annealed in the temperature of 500 °C possessed the highest sensitivity since the annealing at an appropriate temperature could compensate oxygen vacancies to improve the sensing capability. Moreover, with the addition of the SiO_2_ layer, the trapping charges or dangling bonds could be reduced since the thermal grown SiO_2_ layer was near-perfect dielectric with fewer defects.

To investigate the hysteresis effects of the membrane, the tested samples were immersed in buffer solutions with different pH values in an alternate cycle (pH 7, pH 4, pH 7, pH 10, and pH 7). The submerging time was five minutes for each solution. We subjected the above samples to a pH loop of 7→4→7→10→7 over 30 min. The hysteresis voltage was defined as the gate voltage difference between the initial and the terminal voltages measured in the above pH loop. The interior sites of defects could react with the ions existing in the tested solution and thus generate hysteresis response. 

Figure 6a shows the hysteresis voltage of the Sb_2_O_3_ for the as-dep sample and the sample annealed at temperatures from 400 °C, 500 °C, and 600 °C were 47.61 mV, 25.58 mV, 13.82 mV, and 61.98 mV, respectively. Figure 6b shows the hysteresis voltage of the Sb_2_O_3_/SiO_2_ samples for the as-dep sample and the sample annealed at temperatures from 400 °C, 500 °C, and 600 °C were 31.91 mV, 24.42 mV, 12.35 mV, and 36.12 mV, respectively. The Sb_2_O_3_/SiO_2_ sample annealed at a temperature of 500 °C had the lowest hysteresis deviation. The SiO_2_ layer could reduce charge loss and increase the reliability of the interface. Annealing in O_2_ ambient could repair the dangling bonding connection and fill the traps in the film.

To investigate the drift rate of the membrane for long-time reliability, the tested samples were immersed in pH7 buffer solutions and the submerging time was twelve hours. We can use the model of gate voltage drift of pH-ISFET to describe a hopping and/or trap-limited transport mechanism. ‘Drift coefficient’ is a parameter that describes the long-term stability of a pH sensor. The curves of drift effect of the Sb_2_O_3_ and Sb_2_O_3_/SiO_2_ sensing film were measured in pH 7 buffer solution for 12 h as shown in Figure 7a,b respectively. Figure 6a shows the drift rate values of the as-dep Sb_2_O_3_ sample and the Sb_2_O_3_ samples annealed at 400 °C, 500 °C, and 600 °C were 35.43 mV/h, 8.11 mV/h, 7.25 mV/h, and 12.71 mV/h, respectively. Figure 6b shows the drift rate of the as-dep Sb_2_O_3_/SiO_2_ sample and the Sb_2_O_3_/SiO_2_ samples annealed at 400 °C, 500 °C and 600 °C were 12.56 mV/h, 7.15 mV/h, 6.48 mV/h, and 10.31 mV/h, respectively. The Sb_2_O_3_/SiO_2_ sample annealed in O_2_ treatment at a temperature of 500 °C had the lowest hysteresis deviation.

## 3. Conclusions

In this study, double stacked Sb_2_O_3_/SiO_2_ membranes in EIS structures were fabricated. The double stacked membrane annealed at 500 °C had a higher sensitivity, higher linearity, lower hysteresis voltage, and lower drift rate. Multiple material characterizations indicate that double stack structures and annealing could enhance the crystallization and reduce the defects. Sb_2_O_3_/SiO_2_-based with appropriate annealing show promises for future biomedical sensing devices.

## Figures and Tables

**Figure 1 membranes-12-00734-f001:**
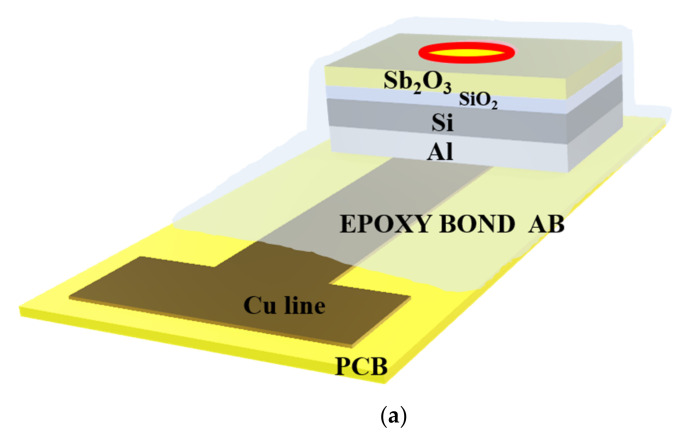
(**a**) The Sb_2_O_3_/SiO_2_ EIS structure. (**b**) The Sb_2_O_3_ sensing membrane applied to the EIS structure with RTA in O_2_ ambient.

**Figure 2 membranes-12-00734-f002:**
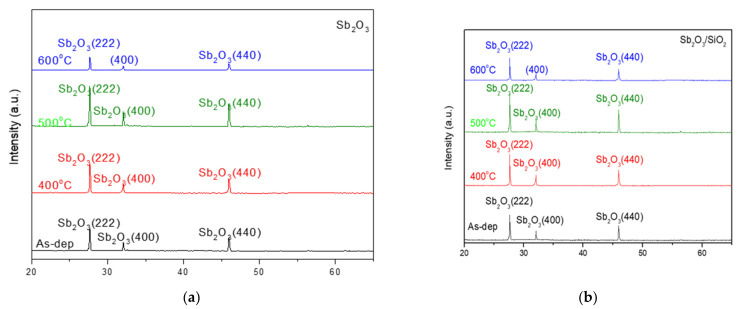
XRD patterns of the (**a**) Sb_2_O_3_ and (**b**) Sb_2_O_3_/SiO_2_ films annealed at various temperatures in O_2_ ambient for 30 s.

**Figure 3 membranes-12-00734-f003:**
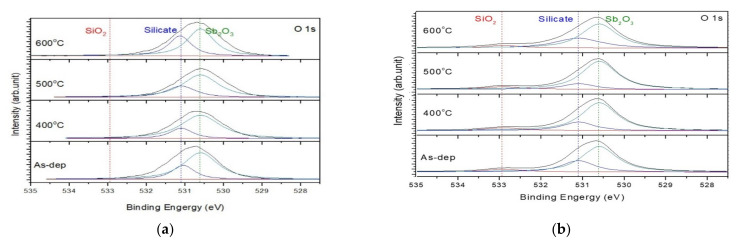
The O 1s XPS results of (**a**) Sb_2_O_3_ film, (**b**) Sb_2_O_3_/SiO_2_ films annealed at various temperatures in O_2_ ambient for 30 s.

**Figure 4 membranes-12-00734-f004:**
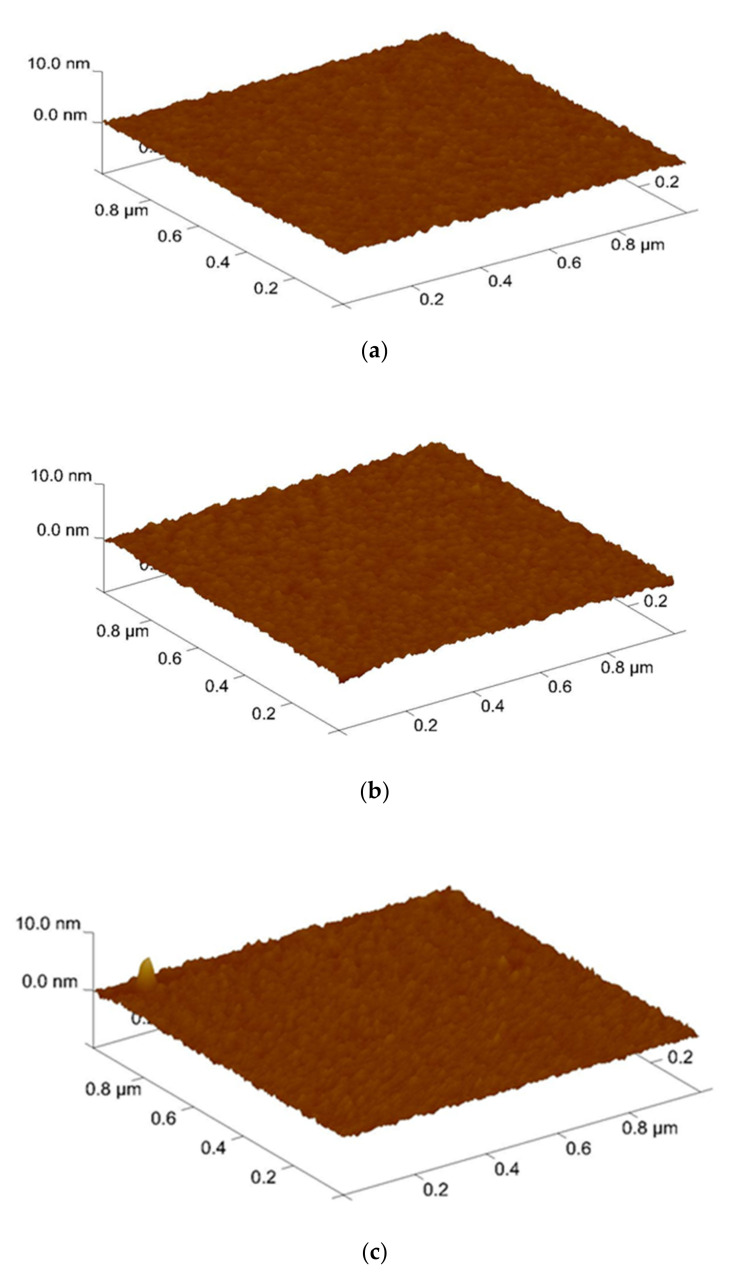
AFM images of Sb_2_O_3_ film film (**a**) As-dep (**b**) 400 °C (**c**) 500 °C (**d**) 600 °C surface after RF sputter in Ar:O_2_ = 20:5 annealing at various temperatures in O_2_ ambient for 30 s. AFM images of Sb_2_O_3_/SiO_2_ film (**e**) As-dep (**f**) 400 °C (**g**) 500 °C (**h**) 600 °C surface after RF sputter in Ar:O_2_ = 20:5 annealing at various temperatures in O_2_ ambient for 30 s.

**Figure 5 membranes-12-00734-f005:**
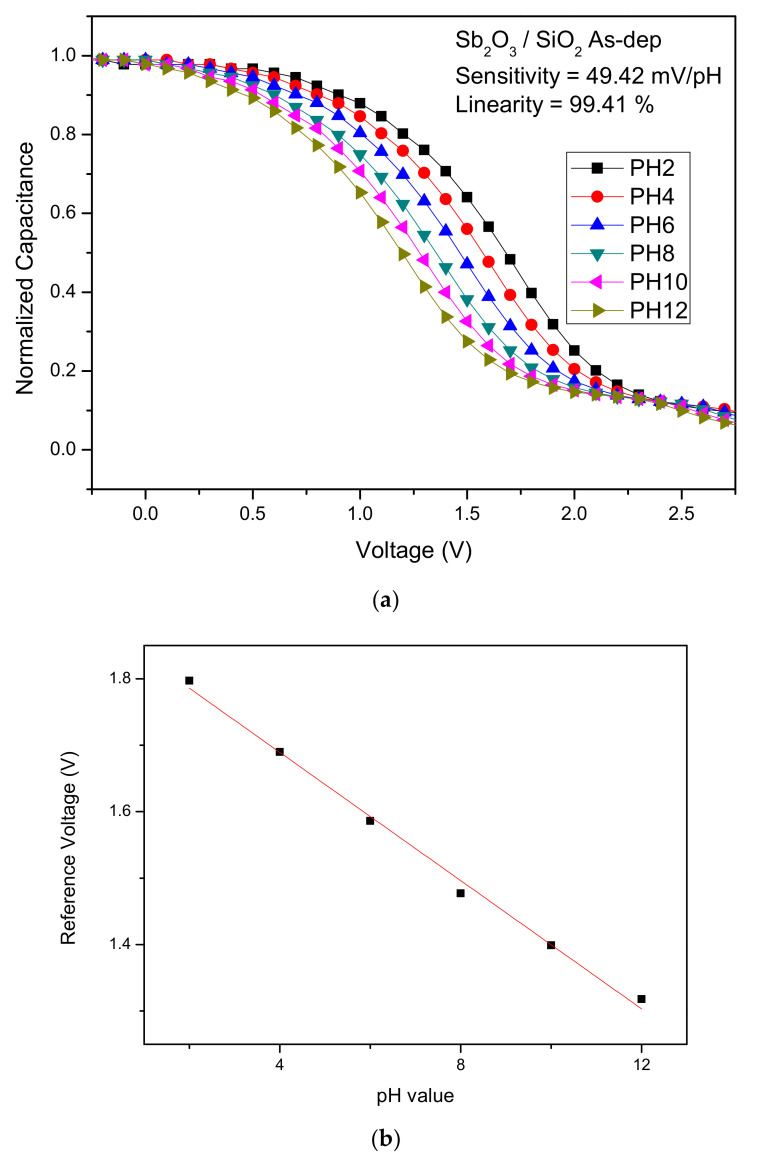
The C-V curves and the extracted sensing data of (**a**) the as-dep Sb_2_O_3_ film, (**b**) the Sb_2_O_3_ films annealed at 500 °C, (**c**) the as-dep Sb_2_O_3_/SiO_2_, and (**d**) the Sb_2_O_3_/SiO_2_, film annealed at 500 °C.

**Figure 6 membranes-12-00734-f006:**
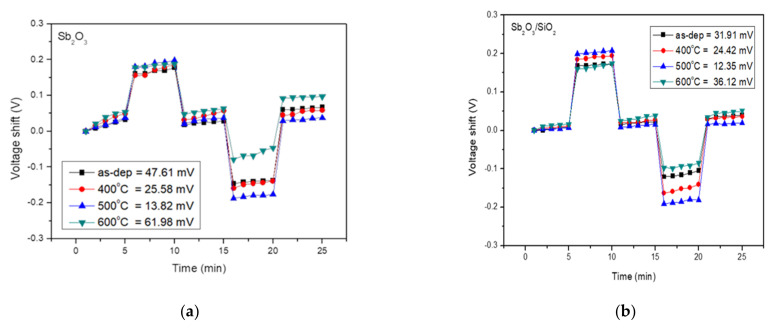
(**a**) Hysteresis of Sb_2_O_3_ sensing membrane with various RTA temperatures in O_2_ ambient during the pH loop of 7→4→7→10→7over a period of 30 min. (**b**) Hysteresis of Sb_2_O_3_/SiO_2_ sensing membrane with various RTA temperatures in O_2_ ambient during the pH loop of 7→4→7→10→7over 30 min.

**Figure 7 membranes-12-00734-f007:**
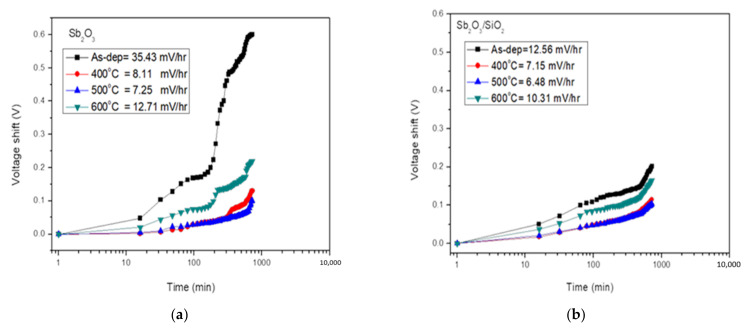
(**a**) Drift voltage of Sb_2_O_3_ sensing membrane annealed with various RTA temperatures in O_2_ ambient, then dipped in pH 7 buffer solution for 12 h. (**b**) Drift voltage of Sb_2_O_3_/SiO_2_ sensing membrane annealed with various RTA temperatures in O_2_ ambient, then dipped in pH 7 buffer solution for 12 h.

## Data Availability

Not applicable.

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
