# Peer review of "Comparison of Sb2O3 and Sb2O3/SiO2 Double Stacked pH Sensing Membrane Applied in Electrolyte-Insulator-Semiconductor Structure"

_membranes, 2022, doi:10.3390/membranes12080734_

Round 1
Reviewer 1 Report
This paper describes results of experimental demonstration of electrolyte-insulator-semiconductor (EIS) capacitors based on Sb2O3/SiO2 bilayer membranes and their characteristics by pH sensing. EIS-type sensors are one of the most effective methods for ion sensing, however, they have some challenges such as selectivity sensitivity and robustness. The research results show that Sb2O3/SiO2 enhances crystallization and grain formation, and strengthens the surface sites of the film, which is a useful finding for industrial applications. It should be considerable to publish in the journal after the following modification as major revision.
- Additional references to the authors' previous publications or references concerning how to prepare the EIS structure
- The authors should provide the reasonableness of the manufacturing process, including the Standard RCA cleaning process and reasoning for conditions. 
- The authors should describe the expected error range for values including the thickness of the SiO2 film.
- The size and possible variation of the sensing area should be described
- Authors should provide details of experimental methods used to evaluate the sensitivity of the solution-based experiments (pH sensitivity, drift, hysteresis) of the sensor. In experiments using solutions, it is impossible to discuss the reliability of the measured values and the reliability of the values if the type of solution (e.g., pH buffer), reagent adjustment method, measurement equipment used, and measurement procedure are unclear.
- Figure 2(a) and (b) differ in vertical and horizontal size and layout (spacing) of each spectrum, which makes comparisons difficult. The layouts in both figures should be the same as possible.
- In the XRD intensities in Figure 2, numerical values of the peak intensities at each temperature relative to as-dep should be given to make comparisons easier.
- Figure 3(a) and (b) have different aspect sizes and layouts (spacing) for each spectrum, making comparisons difficult. The layout of both figures should be the same as possible.
- The sizes of the photographs in Fig. 4(a),(b),(c) and(d) are different, so please align them. Authors should reconsider the need to show Figure 4, as it is difficult to compare them due to insufficient resolution.
- Figure 5 should be further improved in resolution; the accuracy of the AFM measurements should be indicated.
- Authors should explain how the pH sensitivity is calculated for each figure in Figure 6, given the measurement conditions under which the values are collected. Authors should also provide several pH sensitivity values obtained under measurement conditions other than those shown in Figure 6.
- Authors should explain why the values for sensitivity and linearity are valid to the last two decimal places in Figure 6.
- Authors should describe the four pH sensitivities in Figure 6 on one graph to enable comparison of the differences
- The procedure for measuring hysteresis in Figure 7 should be described in more detail. (e.g., with/without co-washing, measurement temperature, humidity, etc.)
- Authors should specify the difference between the first and last voltage at each pH in the hysteresis of Fig. 7.
- Authors should explain why they set pH in the range of 4~10 in the hysteresis evaluation in Fig. 7, which is different from the other experimental results.
- Authors should describe in more detail the experimental procedure for the drift evaluation in Figure 8. (e.g., temperature changes, humidity changes, and other environmental factors measured)
- Authors should describe their evaluation of the pH drift values in Figure 8 compared to other previous studies ( ISFET, SGFET, etc.)
Author Response
Dear reviewer,
Thank you very much for your time. Please see the attached file.

Reviewer 2 Report
In this work the pH sensing membranes based on Sb2O3 and SiO2 were fabricated and were well characterized using microscopic, spectroscopic and electrochemical methods. The results are reliable, however, there are certain issues that need to be cleared.
1. What was the composition of the buffer solutions used in the measurements? Which of the buffer components caused unwanted side reactions with Sb2O3 and SiO2 oxides, generating hysteresis response?
2. The authors need to pay attention to the punctuation and technical quality.
3. Figure 4 is poorly presented, ,, a” and ,,b” need to be added. Second, what is the scale for figure a? Moreover, FESEM were not necessary, as the results don't provide any information related to the sensing performance.
Author Response

(The authors gave the same response as above.)

Round 2
Reviewer 2 Report
The manuscript can be accepted for publication in the present form.